# The Impact of the Macroeconomic Environment on Social Preferences: Evidence from the Global Preference Survey

**DOI:** 10.3390/bs13080648

**Published:** 2023-08-03

**Authors:** Haoyang Li, Xiaomeng Zhang, Shan Jin, Yuanchi Sun, Ding Ma, Cong Wang

**Affiliations:** 1Institute for Advanced Research, Shanghai University of Finance and Economics, Shanghai 200433, China; lihaoyang@mail.shufe.edu.cn; 2Key Laboratory of Mathematical Economics (SUFE), Ministry of Education, Shanghai 200433, China; 3Economics Experimental Laboratory, Nanjing Audit University, Nanjing 210017, China; jinshanecon@gmail.com; 4Gies Business Department, University of Illinois Urbana-Champaign, Champaign, IL 61820, USA; yuanchi2@illinois.edu; 5Department of Economics and Computer Science, Lafayette College, Easton, PA 18042, USA; madi@lafayette.edu; 6School of Economics, Renmin University of China, Beijing 100872, China

**Keywords:** social preferences, trust, altruism, macroeconomics environment, non-linear regression

## Abstract

The effect of social preferences, such as altruism and trust, on economic development is widely recognized. However, the reciprocal impact, i.e., how individuals experience the economic environment and how this shapes their social preferences, has remained largely under-explored. This study sheds light on this reciprocal effect, revealing an intriguing macroeconomic impact on individuals’ social preferences. By harnessing the Global Preference Survey data and a non-linear regression model, our findings highlight an interesting trend: there is a discernible decrease in individuals’ social preference as they experience enhanced economic conditions, and this effect is more pronounced for males. This crucial revelation underscores the importance for researchers and policymakers to take into account the prospective attenuation of social preferences in the pursuit of economic well-being.

## 1. Introduction

For centuries, the field of social sciences has observed human behavior, and the behavior observed has not simply been understood as a function of pure self-interest and rational economic motives alone. This perspective—emphasized by behavioral economists—was even advocated by Adam Smith, who in his seminal work, “The Theory of Moral Sentiments”, delved deeply into the social behaviors of humans and the existence of social preferences [1]. Despite the long-standing tradition of neoclassical economics anchoring its models on self-interest, the resurgence of behavioral economics has drawn the focus back to an exploration of social preferences, lending it renewed significance in the field [2]. Behavioral economists and psychologists argue that human motivation transcends the realm of individual consumption, encapsulating a myriad of social and moral considerations. This is exemplified in altruistic behaviors where individuals assign positive value to the welfare of others [3,4,5,6]. In addition, individuals exhibit concern for fairness in both outcomes [7,8,9] and processes [10,11,12,13], often demonstrating a propensity to share resources even with unknown recipients in the dictator game [14]. The influence of social image concerns is also notable, with individuals seeking to influence perceptions of themselves for both instrumental and hedonic purposes [15,16]. Lastly, people are driven to penalize those involved in misconduct, demonstrating negative reciprocity. Trust, a unique human motivation, underlies economic cooperation. This spectrum of potent motivations is collectively referred to as “social preferences” in the behavioral economics literature.

The importance of social preferences in economic activities is indisputable. Studies by economists and social science researchers have consistently shown that social preferences are significant drivers of economic development. For instance, trust has been found to effectively encourage cooperation [17,18] and influence organizational size [19], both of which are integral to economic development [20,21]. A data-based projection by Algan and Cahuc [18] asserts that if African nations were to achieve Sweden’s level of trust, its per capita income could potentially increase by 5.5 times. Similarly, altruistic behavior can enhance the provision of public goods and social stability, significantly impacting economic development [13,22,23,24,25]. Reciprocity also promotes cooperative behavior and the establishment of social consensus [26,27,28], fostering a benign societal interaction.

In considering the significant role of social preferences in economic development, an intriguing question emerges: What factors shape individuals’ social preferences? What contributes to the emergence of cross-national variations in social preferences? Previous research has attempted to answer these questions and yielded insightful findings. For instance, a nation’s historical memory and culture have been found to influence its inhabitants’ social preferences; in the case of African nations, trust levels have decreased post the slave trade [29,30], while close historical kinship structures have been associated with higher social preferences [31]. In addition, ancestral social preferences appear to have been passed on to future generations [32]. Beyond historical influences, present social conditions also play a role. For example, social identity impacts people’s behavior, leading to increased trust and public goods contributions within shared social identities [33,34]. Education level has a significant correlation with social preference, with higher-education levels tending to increase social preference [35]. Policies and media can also shape social preferences, as exemplified by the improved social preferences resulting from school integration policies [36] or the decreased social preferences in China due to the one-child policy [37,38]. The media’s influence can be seen in the increased social preference for their ethnicity in Rwandan residents who have been exposed to nationalist propaganda [39,40], and increased trust has been observed in women in rural India due to enlightened media education [41]. The degree of freedom in the market economy also affects residents’ social preferences [42]. Neuroeconomists are also actively exploring the causal relationship between the brain and social preferences [43,44].

While it is established that social preferences are shaped by various factors and play a vital role in economic growth and development, whether economic development reciprocally influences and shapes social preferences remains an open question. This is primarily due to data scarcity and methodological limitations. This paper attempts to fill this knowledge gap, following the seminal works [45], and by employing the Global Preference Survey data [35], which contain diverse social preference data from nearly 80,000 residents in 76 countries. In addition, we leverage the nonlinear regression model by Malmendier and Nagel [46], which can effectively control for individual lifetime economic circumstances and weigh memories accordingly. Our study reveals that the economic environment significantly impacts social preferences, with individuals exposed to a better economic environment exhibiting reduced altruism and trust.

Our findings largely align with the conclusions derived from Li and Zhang [45], yet we augment these through the the execution of three comprehensive extensions. Firstly, while existing the study of [45] has merely quantified the correlation between macroeconomic environments and economic preferences (including social preferences), our research employs a predictive methodology through which to establish a causal relationship between macroeconomic experiences and social preferences. This constitutes a significant advancement in the field. Secondly, we deploy a range of model verification techniques to substantiate the legitimacy and necessity of nonlinear regression in this context, bolstering the reliability of our findings. Lastly, we delve further into our analysis by conducting a heterogeneity test. The results suggest that the influence of the macroeconomic environment on women’s social preferences is relatively less potent compared to its impact on men. Moreover, it exerts a smaller impact on the trust of residents in democratic countries as opposed to non-democratic nations. This nuanced understanding of the differential impact of macroeconomic environment factors across various social groups furthers our comprehension of the subject matter, as well as underscores the need for policies to be tailored according to these disparities.

This paper significantly enriches the existing literature in two primary ways. Firstly, it discerns a novel determinant of social preference, namely the macroeconomic environment, thereby broadening our understanding of social preference formation. This research notably pushes the envelope of traditional behavioral economics, which typically views social preferences as inherent and unchanging. Secondly, it posits a bidirectional association between social preference and economic development, emphasizing how a flourishing economy could modulate individuals’ social preferences. These advancements render this study an essential stepping stone toward a more comprehensive comprehension of the intersection between economic dynamics and social behavior.

The remainder of this paper is organized as follows: Section 2 details our methodology and data; Section 3 presents our findings; and Section 4 concludes with a brief discussion.

## 2. Materials and Methods

### 2.1. Empirical Strategy

In this study, we adopted the methodological framework established by Malmendier and Nagel [46] to thoroughly investigate the complex relationship between individual social preferences and their exposure to various macroeconomic environments. A unique aspect of this approach is its capacity to consider the dynamic impact of past experiences, thereby accommodating the potential fluctuation in the influence of distant and more recent events.

We effectively capture the macroeconomic environment of a specific country in a given year by using the logarithm of the real *GDP* per capita metric. Our methodology involves computing a weighted average of each individual’s past macroeconomic environments, with each individual denoted as *i* and residing in country *j*, as articulated in Equation (Equation 1).
(1)$$A_{ij}\left(\lambda \right)=\frac{\sum_{k=1}^{age_{ij}-1}w_{ij}\left(k,\lambda \right)GDP_{j,2012-k}}{\sum_{k=1}^{age_{ij}-1}w_{ij}\left(k,\lambda \right)},\;\mathrm{with}\;w_{ij}\left(k,\lambda \right)={\left(\frac{{\mathrm{age}}_{ij}-k}{age_{ij}}\right)}^{\lambda },$$

Through this refined methodology, we glean a more nuanced understanding of the dynamic interplay between individual social preferences and the encompassing macroeconomic conditions. The term $GDP_{j,2012-k}$ signifies the logarithm of the real *GDP* per capita of country *j* for the year 2012−k, and it is established at constant 2011 national prices (expressed in 2011 USD). Here, *k* functions as a temporal marker—the larger its value, the further the corresponding year is from the survey year. The weight attributed to the *GDP* in the *k*th year before 2012, namely $w_{ij}\left(k,\lambda \right)$, varies based on the individual’s age and the designated parameter λ, which dictates the structure of the weighting function.

The specification of weights is thoughtfully designed to be both parsimonious and flexible, allowing for the weights to either decrease, remain constant, or increase with respect to the temporal distance *k*. The direction of the change is determined by λ, which is directly inferred from the data. In particular, when λ < 0 (>0), a higher emphasis is assigned to more distant (recent) experiences. In the case of λ=0, constant weights are implemented, leading to $A_{ij}$ being a simple average of past *GDP* since the individual’s birth. Once $A_{ij}\left(\lambda \right)$, our main independent variable, is defined, we employ a nonlinear regression model in line with the Malmendier and Nagel [46] framework to quantify the relationship between past macroeconomic environmental experiences and current social preferences. This relationship is modeled by Equation (Equation 2):(2)$$SP_{ij}=\alpha +\beta A_{ij}\left(\lambda \right)+\gamma x_{i}+\tau_{j}+\varepsilon_{ij},$$

In this equation, $SP_{ij}$ represents the social preference scores for individual *i* in country *j* during the survey year. We will delve further into $SP_{ij}$ in Section 2.2, but, generally, a higher $SP_{ij}$ corresponds to increased social preferences, such as higher levels of altruism or trust. $\tau_{j}$ is an array of country fixed effects that captures the effects on social preferences of country-specific time-invariant factors, such as culture and social norms. Finally, given that each individual appears only once in the dataset, we cannot include individual fixed effects. However, to enhance estimation precision, we factor in available data, such as gender and math skills, into $x_{i}$. The nonlinear least square technique was employed to estimate Equation (Equation 2) by searching over the (β, λ) space to minimize the sum of the squared residuals.

### 2.2. Data

The cornerstone of this scholarly paper is the highly esteemed Global Preferences Survey (GPS), which was meticulously compiled by Falk et al. [35]. This survey, with its strong reputation for experimental validation, provides an extensive overview of economic preferences, including attitudes toward risk, time preference, reciprocity, altruism, and trust. The survey, executed across 76 countries in 2012, produced a rich, cross-sectional dataset comprising approximately 80,000 respondents.

Our investigation hones in on social preferences—specifically, altruism, positive reciprocity, negative reciprocity, and trust. We would like to underscore that the GPS applies distinct methodologies to assess these participant preferences. Altruism was evaluated through a self-reported question complemented by a donation game. Positive reciprocity was ascertained through a self-report question and a gift exchange game, while negative reciprocity was measured by multiple self-reporting questions. Trust was gauged via a solitary self-reporting question. This multi-faceted approach involved assigning weights to each question, particularly when multiple questions assess the same preference. This thorough methodology was underpinned by a meticulous experimental validation procedure. Furthermore, these measures, as part of the complete GPS dataset, were standardized to a mean of 0 and a standard deviation of 1. Further insights into the construction of these measures are available in Falk et al. [35]. For a succinct summary of the construction, please refer to Li and Zhang [45]. Demographic information, such as age and gender, was also included in the survey.

Finally, country-level historical real *GDP* per capita and consumption per capita records were drawn from the Penn World Table (PWT) [47]. The final merged dataset encompasses 75 countries, represented by 55,706 individuals. Please note that our analysis exclusively incorporates individuals who were born after *GDP* data became available in their respective countries due to gaps in early *GDP* per capita data in the PWT dataset.

Table 1 serves as a detailed summary of both the initial comprehensive GPS dataset and the final merged dataset used in our study. Within Panel A of the table, we present an inclusive review of the summary statistics for the variables utilized in our analysis. Panel B displays the summary statistics for the primary GPS variables that were not incorporated in our analysis. Panel C presents a comprehensive summary of the demographic information related to the participants. Given the GPS dataset’s representation of a diverse global population, the correlation observed in the summary statistics between the original and final dataset in all three panels substantiates the validity of our final dataset. Finally, Panel D delineates the macroeconomic environment variables.

## 3. Results

### 3.1. Baseline Results

As Table 2 elucidates, the economic circumstances that individuals experienced significantly molded their social preferences.

Positive reciprocity, reflecting the inclination to repay the kindness of others, remains impervious to the macroeconomic environment (p=0.643). Conversely, negative reciprocity, or the predisposition to retaliate against unfair conduct, is significantly influenced by one’s economic experiences (p=0.004). We observe that individuals who have encountered more favorable economic circumstances exhibit a diminished propensity to desire penalizing unfair actions. Remarkably, an individual’s economic conditions significantly impact their altruistic tendencies (p<0.000). Specifically, those who have been exposed to superior economic conditions demonstrate a marked reduction in altruistic behaviors. Trust, a crucial driver of economic development [17,18], was found to be inversely correlated with favorable economic experiences (p=0.001). This intriguing reciprocal relationship might function as an endogenous, spontaneous mechanism preventing the exacerbation of economic inequality, achieved by maintaining a balance between trust-driven economic growth and wealth disparity.

The magnitude of several significant coefficients necessitates closer scrutiny. Our regression analysis reveals that an improvement in an individual’s economic environment, scaled by Euler’s number, corresponds to a decrease in their propensity to wish to retaliate against others or to penalize unfair actions by 0.33 standard deviations. Similarly, an enhancement in the economic environment, when scaled by Euler’s number, results in a 0.29 standard deviation reduction in an individual’s altruistic preferences. The most notable observation is that when the economic circumstances experienced by an individual improves by Euler’s number, there is a decrease in trust by 0.45 standard deviations. In layman’s terms, this means that a decade of consistent rapid growth can lead to a shift in a population’s trust level from above-average to below-average.

Turning attention to control variables, we glean further insights. Age significantly sways social preferences: with advancing age, individuals display an enhanced tendency to reward kindness, a decreased propensity to reprimand unfair behavior, a diminishing inclination toward altruism, and an augmented trust in others. Furthermore, our regression model elucidates on the gender-based disparities in social preferences. We discover that women exhibit a stronger disposition to reciprocate kindness, a reduced readiness to retaliate against unfair actions, and notably higher levels of altruism and trust. These age- and gender-based influences on social preferences align with prior research [48,49,50,51,52], bolstering the robustness of our results.

### 3.2. Regression Diagnostics

In this subsection, we carry out a battery of regression diagnostics to ensure the validity of the estimation strategy. First, we check whether our regression results are driven by potential outliers in the values of the four dependent variables and important covariates. Specifically, we plot the distributions of these variables in Figure 1. Reassuringly, we do not find severeoutliers in these variables.

Second, we test whether the results are affected by potential multicollinearity between the covariates included in the regression. Specifically, we calculate the piecewise correlation between age, math skill, gender, and $A_{ij}$ for each regression. However, since $A_{ij}$ is essentially a nonlinear function of an individual’s age and the *GDP* per capita they experienced across their life course, it will be unknown before we estimate the model. Nonetheless, we compute $A_{ij}$ with the estimated λ and treat it as a known variable. Since λ is different for different dependent variables, $A_{ij}$ also differs by dependent variables, even for the same person. The piecewise correlations reported in Table 3 indicate that there are no economically significant correlations between the key variables in the regression; therefore, our regression results are shielded from the multicollinearity problem.

Lastly, the presence of heteroscedasticity in the error terms may introduce bias into the standard error estimation. In our main regressions, we have clustered the standard errors at the sub-national region level to account for both arbitrary correlations in the error terms of individuals living in the same sub-national regions and the potential heteroscedasticity in the error term. Nonetheless, we also applied the Davidson and MacKinnon (1993 and 2004) [53,54] variance–covariance matrix estimation adjustment in one robustness check, since the two papers cited above show that this adjustment often produces better results when the model is heteroscedastic. The results, shown in Table 4, are very similar to those reported in Table 2, indicating the robustness of our initial estimation results. Given this robustness to heteroscedasticity, we still prefer our initial standard error calculation method, which simultaneously dealt with autocorrelations.

### 3.3. Robustness Check

In order to rigorously examine the robustness of our results, we adopted an alternative approach, deploying the variable of per capita consumption of residents as a surrogate measure for the economic environment. The rationale behind this choice is that residents’ personal consumption level might directly and intuitively influence their perceptions of the economic situation more than *GDP*. The outcomes of our regression analysis, by leveraging this alternative measure, are detailed in Table 5.

Upon meticulous scrutiny of our regression results, it becomes evident that with the switch of the economic environment variable from *GDP* to per capita consumption, the significance of negative reciprocity was considerably diminished. However, the impact on altruistic preference and trust remained robust and statistically significant. This reinforces the strength and credibility of our initial findings.

After incorporating this additional analysis, it fortifies our confidence in concluding that the macroeconomic environment an individual experiences indeed exerts a substantial and meaningful impact on their altruistic preferences and levels of trust. This refined understanding underscores the importance of considering macroeconomic influences when interpreting and predicting individuals’ social preferences, further informing our insights into the nuanced interplay between economic environments and human behavior.

A pertinent question meriting further exploration is the precedence of economic environments and social preferences—does a past economic environment influence people’s social preferences, or is it the other way around? Although our non-linear regression technique and multi-country comparative methodology offer a better understanding of this causality, we have implemented an additional predictive regression for added assurance. This involves regressing individual social preferences against the future economic landscape of the individual’s resident country. If a significant relationship between social preferences and future economy emerges, and is consistent with the correlation observed with past economic conditions, it becomes challenging for us to decipher the causal relationship. However, if a variation is noted—for instance, if a past economic environment resulted in increased distrust (as shown in our primary results) and trust is positively linked with the future economic scenario—we can infer a more substantial conclusion. This could suggest that a positive past economic environment cultivated the distrust, rather than such distrust fostering an improved economic environment.

Table 6 presents a noteworthy observation, i.e., that there exists a positive correlation between positive reciprocity and future economic growth. This suggests that nations with high positive reciprocity are likely to experience enhanced economic development in the future. However, our primary results indicate no significant relationship between a past economic environment and positive positive reciprocity. Negative positive reciprocity, on the other hand, exhibits a contrasting pattern. Current negative positive reciprocity of a country’s residents does not influence future economic growth, yet our main findings reveal a substantial connection with the past economic conditions experienced by these residents. Our principal results suggest that adverse past economic conditions can lead individuals toward increased altruism, but interestingly, such selflessness does not predict future economic conditions. Most intriguingly, a significant positive correlation exists between the current trust preferences of residents and future economic development, suggesting that the higher the level of trust among residents, the better the future economic development—a finding that is consistent with prior research [55,56]. However, we have discovered that past economic environment experiences has a negative correlation with an individual’s trust preference, indicating that exposure to a superior economic environment can reduce trust preference. This might be an endogenous regression of development, thereby aiding in the reduction in development inequality.

In conclusion, our core findings illustrate a discrepancy between the influence of past economic environments on social preferences and the predictability of social preferences on future economic situations. This substantiates the initial query regarding causality, affirming that our investigation establishes causation rather than mere correlation.

### 3.4. Heterogeneous Analysis

We further extend the baseline analyses in Section 3.1 to examine how the estimated relationship between social preference and macroeconomic conditions varies across individuals and countries with different characteristics. Building upon prior research and the primary outcomes of our investigation, it was discerned that there exist marked disparities in the social preferences between women and men. Such a phenomenon piqued our curiosity and prompted us to explore if the influence of the macroeconomic environment on social preferences exhibits any gender differences. To address this concern, we augmented our basic regression with an interaction term for gender and a weighted economic environment, $A_{ij}\left(\lambda \right)$, which was performed to examine the presence and extent of gender disparities in the observed effects. The outcomes of this nuanced examination are documented in Table 7.

A careful inspection of Table 7 reveals that there are indeed appreciable gender differences concerning the impact of the macroeconomic environment on social preferences. For instance, while the macroeconomic environment does not significantly affect men’s predisposition toward positive reciprocity, it does play a considerable role for women. Specifically, a more favorable economic milieu significantly heightens women’s inclination to reciprocate in others’ benefits. In terms of negative reciprocity, our analysis did not reveal any noteworthy gender-specific disparities. However, when it comes to trust and altruistic behavior, the gender dynamics become more significant. While a superior economic environment significantly dampens both altruistic tendencies and trust levels in the general population, this impact is markedly more attenuated in women when compared to men.

It is crucial to mention here that our study does not merely identify these gender-based discrepancies in response to the macroeconomic environment; it also invites us to reflect on their broader societal and economic implications. The subtle, yet significant, ways in which gender mediates the relationship between the macroeconomic environment and social preferences offer valuable insights. These insights could inform future research and policy decisions, enriching our understanding of the dynamics of social preference formation in varying economic circumstances, thereby advancing this vital field of study.

In our subsequent heterogeneity analysis, we cast a spotlight on the unique political systems of each studied nation. We enhanced the original regression model by incorporating an interaction term between a country’s Democracy Index [57] and its macroeconomic experience. The results of this nuanced analysis are visually represented in Table 8.

A closer examination reveals a significantly positive interaction between the Democracy Index and the economic environment when it comes to trust (notwithstanding its primary negative effect). This implies that, in a more democratic polity, the negative impact of a robust economic environment on reducing trust is diminished. When considered in conjunction with our previous discussions regarding the ’trust-trap’ theory of economic development, a theory asserting that, while economic development requires trust, an improved economy paradoxically reduces trust and thus hampers development; thus, democracies appear to be less vulnerable than non-democratic countries. This observed phenomenon suggests that encouraging political democratization could be a potent strategy in mitigating the detrimental effects of the trust trap. This study thereby underscores the intricate interplay between political structures and economic realities, suggesting the need for integrated, multi-faceted policy approaches.

## 4. Discussions

This investigation has utilized a widely used and validated data source [58,59,60]: the Global Preference Survey. As per the most recent statistics derived from Google Scholar, the dataset in question has been referenced over a thousand times, and this widespread citation not only underscores its value in the research community, but also provides further corroboration of the data’s validity. This data source was used in conjunction with a sophisticated nonlinear regression model to evaluate the profound impact of macroeconomic conditions on individual social preferences. The findings illuminate a compelling inverse relationship: the better the economic environment an individual encounters, the less altruistic they tend to be, and the lower their trust level typically drops.

Our findings align with and arguably broaden the validity of several existing hypotheses and the results presented in the academic literature. For instance, a study by Piff et al. (2018) established a correlation between higher social status (which is often associated with superior economic circumstances) and diminished ethical behavior coupled with heightened self-interest [61]. The authors theorized that wealthier individuals may be less exposed to societal pressures that foster trust and cooperation, leading them to display reduced altruistic tendencies. This viewpoint is echoed by Smith (2007), who found a negative correlation between socio-economic status and altruistic behavior [62]. Similarly, Li and Zhang (2023) identified this pattern while investigating broader economic preferences [45].

Conversely, there is a body of research proposing that prosperous economic conditions might engender higher levels of trust and altruism. Eriksson and Simpson (2014) found a positive association between economic security and interpersonal trust, indicating that a stable economy could boost trust among its populace [63]. However, it should be noted that the positive economic environment referred to in their study primarily pertains to stability, representing psychological expectations of the populace, rather than overarching changes spurred by economic growth. Hence, it does not contradict our findings.

The implications of these insights are manifold and hold considerable significance for our understanding of human behavior within a macroeconomic context. This research elucidates that individual social preferences are not static but evolve in response to the broader economic environment. An individual’s altruism and trust are revealed to be not merely personal traits but complex adaptive responses to economic circumstances. As such, this study significantly extends the boundaries of existing research on behavioral economics, which typically perceives social preferences as innate and invariable.

From a policy perspective, our findings shed light on the potential effects of macroeconomic policies on societal behaviors. Policymakers might need to consider the implications of economic policies on social preferences, especially trust and altruism, and how these changes could, in turn, influence the economy and societal dynamics. For instance, understanding that improved economic conditions may diminish altruistic tendencies could guide policies that encourage charitable giving and social responsibility. Policymakers might need to devise strategic incentives to maintain or enhance altruism, thereby promoting societal well-being, in periods of economic prosperity. Similarly, the revelation that heightened economic conditions may reduce trust levels warrants attention. Trust is a fundamental building block of economic transactions and social cooperation [64,65,66], and a decline in trust could have far-reaching impacts on the economy. Policymakers might need to implement initiatives that foster trust, such as enhancing transparency in financial transactions, reinforcing legal protections, and promoting ethical business practices. Moreover, our findings might also bear implications for workplace policies and corporate culture. If economic prosperity may reduce altruism and trust, businesses may need to invest more in fostering a culture of mutual support, cooperation, and trust, especially in times of economic growth. A salient finding from our research also offers strategic guidance for policy interventions. Within a more democratic system, a prosperous economic environment can incite a heightened aversion to inequity. Based on this conclusion, we propose specific policy recommendations. If the aspiration is to enhance the economy’s sway over residents’ preferences, (an influence typically producing beneficial outcomes as exemplified by the trust mechanism previously discussed, which can efficaciously curtail international inequality), then there should be concerted efforts to foster a government that is more democratic.

## 5. Conclusions

In conclusion, this study has unveiled a novel dimension of how macroeconomic conditions influence individual social preferences. The insights offer a fresh perspective on the dynamic interplay between economic environments and human behaviors, suggesting that an individual’s social preferences are a product of both their personal characteristics and their economic experiences. It is our hope that these findings will prompt further investigation into this underexplored territory and inform more nuanced and effective economic and social policies.

## Figures and Tables

**Figure 1 behavsci-13-00648-f001:**
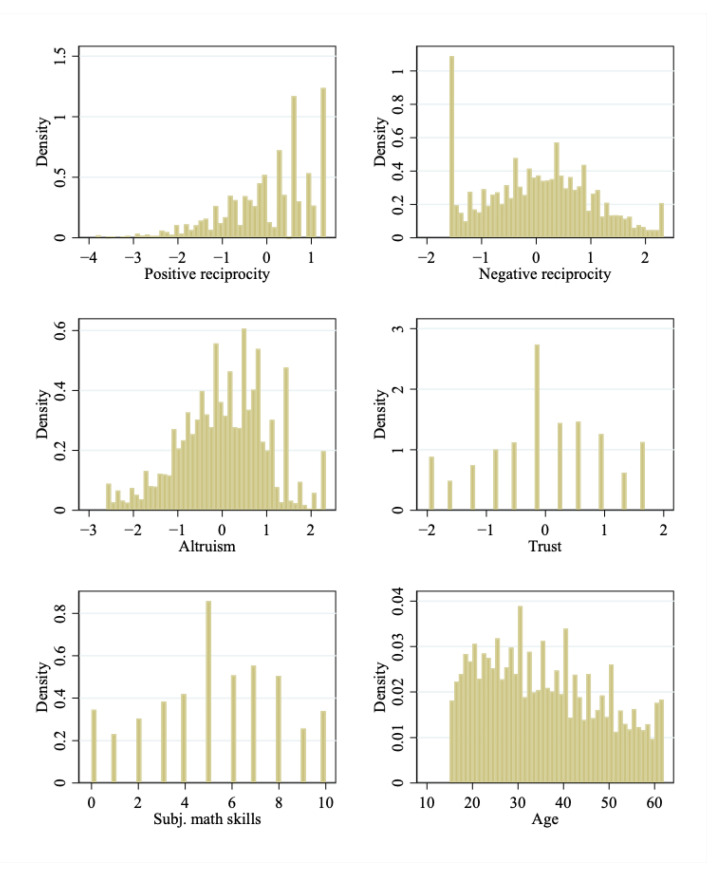
Distributions of Key Variables in the Regressions.

**Table 1 behavsci-13-00648-t001:** Summary Statistics.

VARIABLES	The Original GSP Dataset			The Final Dataset		
	N	Mean	Std. dev.	N	Mean	Std. dev.
*Panel A: Social Preferences Used in the Paper*						
Positive reciprocity	80,189	0.000	1	55,879	−0.003	1.008
Negative reciprocity	78,536	0.000	1	55,257	0.038	0.992
Altruism	79,903	0.000	1	55,767	0.033	0.980
Trust	78,774	0.000	1	55,291	−0.020	1.000
*Panel B: Preferences Not Used in the Paper*						
Risk-taking score	79,703	0.000	1	55,706	0.106	0.978
Patience	79,730	0.000	1	55,732	0.039	1.010
*Panel C: Demographics*						
Age	80,061	41.815	17.492	55,706	35.077	12.845
Female	80,337	0.547	0.498	55,706	0.539	0.498
Math skill	79,211	5.176	2.825	55,706	5.226	2.791
*Panel D: Macroeconomic Environment*						
Average historical log *GDP* per capita				55,706	12.582	1.723
Average historical log consumption				55,706	12.166	1.624

Note: Panel A reports the summary statistics for the variables of social preferences used in this paper. We show the summary statistics for both the original full GSP dataset and the final merged dataset. Positive/negative reciprocity, altruism, and trust were all standardized across the full GSP sample to have mean 0 and standard deviation 1. Panel B reports the summary statistics for the variables of other economic preferences that were not used in this paper. Panel C reports the summary statistics for the variables of demographics used in this paper. Panel D reports the summary statistics for the variables of macroeconomic environments used in this paper.

**Table 2 behavsci-13-00648-t002:** Effect of Past Macroeconomic Environment Experience on Social Preferences.

VARIABLES	Macroeconomic Environment: Logged *GDP* per Capita			
	(1)	(2)	(3)	(4)
	Pos. Recip.	Neg. Recip.	Altruism	Trust
A	0.028	−0.328 ***	−0.286 ***	−0.448 ***
	(0.061)	(0.114)	(0.079)	(0.140)
λ	0.000	1.135 **	0.501 *	1.960 **
	(1.455)	(0.511)	(0.273)	(0.890)
Age	0.002 **	−0.009 ***	−0.002 ***	0.002 **
	(0.001)	(0.001)	(0.001)	(0.001)
Math Skill	0.037 ***	0.040***	0.042 ***	0.057 ***
	(0.003)	(0.003)	(0.003)	(0.003)
Female	0.042 ***	−0.109 ***	0.100 ***	0.061 ***
	(0.010)	(0.010)	(0.010)	(0.011)
Experience weighted	Yes	Yes	Yes	Yes
Country fixed effects	Yes	Yes	Yes	Yes
Number of observations	54,235	53,634	53,680	55,178
Adjusted $R^{2}$	0.141	0.095	0.127	0.118

Note: This table reports the estimation results for effect of macroeconomic environments on social preferences. One’s past macroeconomic environment experiences are measured by the historical real *GDP* per capita of the country they live in. Columns (1) to (4) each report one economic preference result, respectively. Standard errors clustered at the sub-national region level are reported in the parentheses. ***, **, and * denote statistical significance at the 1%, 5%, and 10% levels, respectively.

**Table 3 behavsci-13-00648-t003:** Piecewise Correlation between Key Variables in the Regressions.

VARIABLES	(1)	(2)	(3)	(4)	(5)	(6)
	A_Pos. Recip.	A_Neg. Recip.	A_Altruism	A_Trust	Age	Math
Age	0.0546	0.1516	0.1139	0.174		
Math	0.1738	0.1685	0.171	0.1665	−0.032	
Female	0.0191	0.0213	0.0204	0.0219	0.0143	−0.1149

**Table 4 behavsci-13-00648-t004:** Alternative Standard Error Calculation Method that Works Better under Heteroscedasticity.

VARIABLES	(1)	(2)	(3)	(4)
	Pos. Recip.	Neg. Recip.	Altruism	Trust
A	0.028	−0.328 ***	−0.286 ***	−0.448 ***
	(−0.047)	(−0.092)	(−0.073)	(−0.114)
λ	0.000	1.135 **	0.507 *	1.960 ***
	(−1.329)	(−0.444)	(−0.262)	(−0.717)
Age	0.001***	−0.009 ***	−0.002 ***	0.002 **
	(−0.001)	(-0.001)	(0.001)	(−0.001)
Math skill	0.037 ***	0.040 ***	0.042 ***	0.057 ***
	(−0.002)	(−0.002)	(−0.002)	(−0.002)
Female	0.042 ***	−0.109 ***	0.100 ***	0.061 ***
	(−0.008)	(−0.008)	(−0.008)	(−0.008)
Number of observations	54,235	53,634	53,680	55,291
Adjusted $R^{2}$	0.140	0.095	0.126	0.118

Note: ***, **, and * denote statistical significance at the 1%, 5%, and 10% levels, respectively.

**Table 5 behavsci-13-00648-t005:** Robustness Check: Using Per Capita Consumption as a Macroeconomic Indicator.

VARIABLES	Macroeconomic Environment: Logged Consumption per Capita			
	(1)	(2)	(3)	(4)
	Pos. Recip.	Neg. Recip.	Altruism	Trust
A	0.011	0.073	−0.193 ***	−0.413 ***
	(0.070)	(0.095)	(0.068)	(0.094)
λ	0.121	1.221	0.375	1.482 ***
	(4.651)	(1.992)	(0.232)	(0.423)
Age	0.001	−0.006 ***	−0.002 ***	0.001
	(0.001)	(0.001)	(0.001)	(0.001)
Math skill	0.037 ***	0.040 ***	0.042 ***	0.057 ***
	(0.003)	(0.003)	(0.003)	(0.003)
Female	0.042 ***	−0.110 ***	0.100 ***	0.061 ***
	(0.010)	(0.010)	(0.010)	(0.011)
Experience weighted	Yes	Yes	Yes	Yes
Country fixed effects	Yes	Yes	Yes	Yes
Number of observations	54,235	53,634	53,680	55,291
Adjusted $R^{2}$	0.141	0.094	0.127	0.118

Note: This table reports the estimation results for the effect of the macroeconomic environment on economic preferences. One’s past macroeconomic environment experiences are measured by the historical real consumption per capita of the country where they live in. Columns (1) to (4) each report one economic preference result, respectively. Standard errors clustered at the sub-national region level are reported in the parentheses. *** denote statistical significance at the 1% levels, respectively.

**Table 6 behavsci-13-00648-t006:** Robustness Check: What Came First.

VARIABLES	(1)	(2)	(3)	(4)
	Macroeconomic Environment: Logged future *GDP* per Capita (2013–2017)			
Pos. Recip.	0.053 ***			
	(0.017)			
Neg. Recip.		0.102 ***		
		(0.018)		
Altruism			−0.002	
			(0.016)	
Trust				0.104 ***
				(0.020)
Age	0.019 ***	0.020 ***	0.020 ***	0.019 ***
	(0.002)	(0.002)	(0.002)	(0.001)
Math skill	0.056 ***	0.054 ***	0.058 ***	0.051 ***
	(0.005)	(0.005)	(0.005)	(0.005)
Female	0.073 ***	0.087 ***	0.076 ***	0.071 ***
	(0.013)	(0.013)	(0.013)	(0.013)
Number of observations	55,764	55,156	55,178	55,291
Adjusted $R^{2}$	0.091	0.098	0.088	0.098

Note: This table reports the estimation results for the effect of economic preferences on future economic environment. One country’s future macroeconomic environment is measured by the average real consumption per capita in 2013–2017 of the country where the person lives in. Columns (1) to (4) each report one economic preference result, respectively. Standard errors clustered at the sub-national region level are reported in the parentheses. *** denote statistical significance at the 1% levels, respectively.

**Table 7 behavsci-13-00648-t007:** Heterogeneous Analysis: Gender Differences.

VARIABLES	Macroeconomic Environment: Logged *GDP* per Capita			
	(1)	(2)	(3)	(4)
	Pos. Recip.	Neg. Recip.	Altruism	Trust
A	−0.001	−0.330 ***	−0.332 ***	−0.499 ***
	(0.029)	(0.115)	(0.081)	(0.148)
A × Female	0.034 ***	−0.007	0.061 ***	0.043 ***
	(0.010)	(0.009)	(0.010)	(0.009)
λ	−0.315	1.172 **	0.548 **	2.173 **
	(0.566)	(0.524)	(0.277)	(0.983)
Age	0.001 **	−0.009 ***	−0.002 ***	0.001**
	(0.001)	(0.001)	(0.001)	(0.001)
Math skill	0.037 ***	0.040 ***	0.042 ***	0.057 ***
	(0.003)	(0.003)	(0.003)	(0.003)
Female	−0.259 ***	−0.048	−0.445 ***	−0.333 ***
	(0.090)	(0.083)	(0.087)	(0.090)
Experience weighted	Yes	Yes	Yes	Yes
Country fixed effects	Yes	Yes	Yes	Yes
Number of observations	54,235	53,634	54,130	53,680
Adjusted $R^{2}$	0.142	0.095	0.129	0.118

Note: This table reports the estimation results for the effect of the macroeconomic environment on economic preferences. One’s past macroeconomic environment experiences are measured by the historical real *GDP* per capita of the country where they live in. Columns (1) to (4) each report one economic preference result, respectively. Standard errors clustered at the sub-national region level are reported in the parentheses. *** and ** denote statistical significance at the 1% and 5% levels, respectively.

**Table 8 behavsci-13-00648-t008:** Heterogeneous Analysis: Democracy.

VARIABLES	Macroeconomic Environment: Logged *GDP* per Capita			
	(1)	(2)	(3)	(4)
	Pos. Recip.	Neg. Recip.	Altruism	Trust
A	0.179	−0.400 ***	−0.328 ***	−0.718 ***
	(0.109)	(0.128)	(0.111)	(0.227)
A × Demo	−0.023	0.020	0.008	0.056 *
	(0.014)	(0.020)	(0.016)	(0.030)
λ	−0.114	0.865	0.474	2.146
	(0.585)	(0.555)	(0.292)	(1.409)
Age	0.002 **	−0.009 ***	−0.002 ***	0.002 **
	(0.001)	(0.001)	(0.001)	(0.001)
Math skill	0.037 ***	0.040 ***	0.042 ***	0.057 ***
	(0.002)	(0.003)	(0.003)	(0.003)
Female	0.041 ***	−0.109 ***	0.100 ***	0.061 ***
	(0.010)	(0.010)	(0.010)	(0.011)
Demo	−0.089 *	−193.074 ***	−19.943 ***	−96.474 ***
	(0.047)	(0.188)	(0.123)	(0.266)
Experience weighted	Yes	Yes	Yes	Yes
Country fixed effects	Yes	Yes	Yes	Yes
Number of observations	54,235	53,634	54,130	53,680
Adjusted $R^{2}$	0.141	0.095	0.127	0.118

Note: This table reports the estimation results for the effect of the macroeconomic environment on economic preferences. One’s past macroeconomic environment experiences are measured by the historical real *GDP* per capita of the country where they live in. Columns (1) to (4) each report one economic preference result, respectively. Standard errors clustered at the sub-national region level are reported in the parentheses. ***, **, and * denote statistical significance at the 1%, 5%, and 10% levels, respectively.

## Data Availability

Not applicable.

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
