# Peer review of "The Impact of the Macroeconomic Environment on Social Preferences: Evidence from the Global Preference Survey"

_behavsci, 2023, doi:10.3390/bs13080648_

Round 1

Reviewer 1 Report

The article discusses how economic growth, that is, better living conditions, impact social preferences, which makes the article interesting and relevant, as research generally assesses how preferences impact economic development.
The theme is relevant and brings important news, adds value to the literature and presents innovations for applied research.
The authors present consistent conclusions correlated with the objective of the research.

The article has merits for publication, but it should expand the discussion with citation of more authors, who support the results and who refute the results. For this, it is necessary to deepen the analysis so that it can dialogue with other specific authors in the area. For the other 3 or 4 authors who discuss their main results.

Reviewer 2 Report

The paper presents a research study that explores the relationship between macroeconomic conditions and individual social preferences. The study utilizes the Global Preference Survey (GPS) dataset, which includes information from approximately 80000 respondents across 76 countries in 2012, whereby 55706 were retained for further analysis. The dataset measures social preferences such as altruism, positive reciprocity, negative reciprocity, and trust using self-reported questions and game-based tasks. The historical real GDP per capita data from the Penn World Table is incorporated to provide macroeconomic context. The authors build on Falk et al. (2018) and reverse the relationships' directions.

The methodology employed in the study is typical for the field and involves assigning weights to each question and standardizing the measures to account for variations. The researchers also use country fixed affects and control for demographic factors age, gender, and mathematical proficiency in their nonlinear regression analysis.

Furthermore, the authors apply sensitivity analysis of the results by using an alternative measure of macroeconomic conditions, per capita consumption. The analysis confirms the robustness of the initial findings regarding the impact of economic conditions on altruism and trust.

Heterogeneity analysis explores gender differences in the relationship between macroeconomic conditions and social preferences. The results indicate that the effects of the macroeconomic environment on social preferences differ between men and women. However, the heterogeneity has not been examined for other control variables, like math skill.

The main results of the analysis reveal interesting insights. While a reverse direction of the relationships has been found in previous research, this study finds that individuals' economic circumstances significantly shape their social preferences (particularly altruism and trust).

I do not detect any issues with the language. The language used in the paper is formal and academic in nature. The writing style is objective and focused on presenting the research findings and methodology, allowing readers to understand and evaluate the findings.

A few comments to improve the paper:

Since the relationships stand in both directions, a question still remains - "what came first", social preferences or macroeconomic environment? It seems that further study is needed to capture mechanisms relevant for any intervention and policy making. Authors are encouraged to defy this in their discussion.

The fist line of discussion claims that an innovative data source (Global Preference Survey) has been used, but a quick overview shows almost a 1000 references with published papers that have been using the same dataset: https://www.briq-institute.org/global-preferences/publications . The rest of the paper is devoid of sensationalism and this part should be corrected accordingly.

In addition, the authors should discuss the effect size (not just coefficient significance) and determination coefficient size when reporting the results. This would underscore the comprehensive nature of the observed relationships.

Also, the standard econometric model checking is missing (residuals analysis - heteroscedasticity, nonlinearity, outliers; goodness-of-fit measures, overall model significance, multicollinearity) . The robustness has been checked via sensitivity analysis using replacement variable for GDP per capita, but we still don't know if the model satisfies the usual assumptions. If the authors do not want to burden the paper with technical details, they can place that in the appendix or supplementary file. However, conducting these checks is crucial for ensuring the scientific robustness and reliability of the results.

Round 2

Reviewer 2 Report

The authors have made a substantial improvement to their paper.

However, I am concerned about the scatter plots shown in the new Figures 2 and 3. The shape of the scatter plots (the arrangement of the dots) raises doubts about the suitability of linear relationships, if any. Ideally, the scatter plots used in the main model assessment should have indicated the necessity of using nonlinear regression (and I sincerely hope the authors have checked that). However, the additional analysis presented in Figures 2 and 3 suggests that the analysis in the last part of the chapter 3.4. was improperly employed and should be disregarded, along with any conclusions that stem from that analysis.

The rest of the paper is improved and I think that the paper can be published after the removal of the improper analysis (related to Figure 2 and 3) and related conclusions.
